# Development and Evaluation of Nutritional and Quality Standard of Beef Burger Supplemented with Pumpkin (*Cucurbita moschata*) Seed Flour

**DOI:** 10.3390/foods13111702

**Published:** 2024-05-29

**Authors:** Flávia Alexsandra B. Rolim de Melo, Maria Brígida Fonseca Galvão, Antônio Félix da Costa, Carla Fabiana da Silva, Jenyffer Medeiros Campos Guerra, Thayza Christina Montenegro Stamford

**Affiliations:** 1Programa de Pós-graduação de Nutrição, Universidade Federal de Pernambuco, Av. Profª Morais Rego, 1235, Cidade Universitária, Recife CEP 50670-901, Brazil; flavia.rolim@ufpe.br (F.A.B.R.d.M.); maria.brigida@ufpe.br (M.B.F.G.); jenyffer.campos@ufpe.br (J.M.C.G.); 2Laboratório de Microbiologia Aplicada-LaMAp, Centro de Ciências Médicas, Universidade Federal de Pernambuco, Av. Profª Morais Rego, 1235, Cidade Universitária, Recife CEP 50670-901, Brazil; 3Instituto Agronômico de Pernambuco, Av. General San Martin, 1371, Bongi, Recife CEP 50761-000, Brazil; felix.antonio@ipa.br; 4Departamento de Engenharia Química, Universidade Federal de Pernambuco, Av. Profª Morais Rego, 1235, Cidade Universitária, Recife CEP 50670-901, Brazil; carla.fsilva@ufpe.br

**Keywords:** residue use, meat products, functional foods

## Abstract

The composition of pumpkin seeds includes bioactive compounds, proteins, polyunsaturated fatty acids, and dietary fibers. Thus, the objective of this research was to develop and evaluate the nutritional and quality standard of beef burgers supplemented with pumpkin seeds (*Cucurbita moschata*) added in different proportions. To process the pumpkin seed flour (PSF), the seeds were sanitized, dried in an oven, crushed, and sieved. Through such means, three formulations of beef burgers were prepared, named S (without the addition of PSF), F5 (with the addition of 5% (*w*/*w*) of PSF), and F10 (with the addition of 10% (*w*/*w*) of PSF). The respective results for burgers P, F5, and F10 were as follows (*w*/*w*): proteins 17.61%, 18.04%, 19.86%; lipids 12.19%, 12.42%, 14.55%; ash 1.77%, 1.86%, 1.94%; fibers 0%, 0.88%, 1.76%; phenolic compounds 39.55, 82.93, 90.30 (mg/g); and total antioxidant capacity 11.09%, 18.48%, 24.45%. Regarding the sensory analysis attributes, tasters gave sample F10 scores lower than 7. However, the standard and F5 samples showed results higher than 7 for all parameters. For the determination of shelf life, an expiration date of 30 days was established. It was observed that adding PSF to industrialized products adds nutritional value with the inclusion of polyunsaturated fats, phenolic compounds, and dietary fibers.

## 1. Introduction

*Cucurbita moschata* L. has great economic and social relevance, being a widely consumed fruit, with great acceptance for its sensorial and nutritional characteristics [1]. The consumption of pumpkin is considered beneficial to health due to its composition of essential vitamins, being a source of minerals, proteins, as well as poly- and monounsaturated fatty acids, with a majority of linoleic, oleic, palmitic, and stearic acids, in addition to being a source of fiber [2,3]. Pumpkin seeds are rich in phenolic and antioxidant compounds, such as squalene, phytosterols, tocopherols (α, β, γ, and δ), tocotrienols, carotenoids, and flavonoids [4,5]. They are also a valuable source of proteins (24–40%) and lipids (37–46%), with a predominance of linoleic (42.6–46.3%), oleic (18–30%), stearic (9.5–18.7%), and palmitic (12.3–18.1%) fatty acids, minerals (magnesium, phosphorus, potassium, sodium, calcium, iron, copper, manganese, zinc, and selenium), and fiber (10–25%) [4]. Based on their composition, pumpkin seeds demonstrate nutraceutical properties [1] and can be used as a dietary supplement to help curb deficiency disorders, as well as to prevent cardiovascular diseases [6,7,8].

A diet enriched with pumpkin seeds brings significant benefits, such as reduced serum triglyceride levels, increased fecal excretion of cholesterol, and decreased lipid accumulation in hepatocytes [9]. Nonetheless, pumpkin seeds feature the same antinutritional factors that may restrict their application in food formulations, such as phytate, oxalate, hydrocyanic acid, and nitrate [2]. These compounds can diminish protein digestibility and amino acid bioavailability, consequently impacting the protein’s overall quality. Furthermore, the elevated levels of phytic acid in pumpkin seeds may exacerbate these antinutritional effects by binding to minerals such as zinc, iron, and calcium, thereby reducing their absorption potential. However, the utilization of traditional methods, such as heat treatment, soaking, sprouting, and fermentation, can significantly reduce the percentage of these compounds [4].

Pumpkin seed flour (PSF) is typically utilized for enriching soups, cookies, pancakes, and breads [2]. There is currently a growing demand for healthier foods, particularly in the meat products sector. The presence of excessive amounts of sodium, saturated fats, cholesterol, and synthetic additives in meat products contributes to a higher risk of metabolic diseases. Thus, there is a need to produce functional and healthy foods capable of imbuing meat products with benefits that will positively impact consumer health [3,10].

Beef burgers are considered one of the most commercialized and consumed frozen meat products by all social classes because they are easy to prepare and have attractive sensory attributes [10]. However, despite meat being rich in nutrients, a source of proteins, and containing all essential amino acids, there is evidence that high consumption of processed meat products contributes to the establishment of cardiovascular diseases, type 2 diabetes, obesity, and some types of cancer. In response, the consumption of meat products has been decreasing due to growing consumer concerns about their effect on human health and the environment. In order to associate meat products with functional claims that can promote health benefits, research has been developed to improve these products by associating them with vegetables. This association, in addition to contributing to the reduction in production costs, suggests the use of bioactive compounds that can improve the nutritional quality of such products [3]. In this context, supplementing beef burgers with PSF could add nutritional, functional, and commercial value to the product. 

Therefore, this research aims to develop and evaluate the nutritional and quality standard of beef burgers supplemented with *Cucurbita moschata* seeds added in different proportions. It is believed that PSF, due to the presence of bioactive compounds (already recognized in the literature), may add functional claims to the innovative product prepared, which may bring benefits to consumer health and assist in the promotion of technological and sustainable actions.

## 2. Materials and Methods

### 2.1. Reception of Raw Materials

Pumpkin seeds (*C. moschata* L.) were collected from merchants in the metropolitan region of Recife, Pernambuco, Brazil, and selected according to the species’ morphological characteristics, such as white color, flat, oval shapes, green color inside, and surfaces free of mechanical damage and visible fungal infections. The pumpkins (*C. moschata* L.) were grown in an integrated and sustainable agroecological production system by family farmers; the fruits were harvested 60 days after anthesis; and cooled ground beef (chuck) and the dry ingredients used to make the burgers were obtained from local commerce in Recife, Pernambuco, Brazil.

### 2.2. Obtaining the Flour (PSF)

In the production of PSF, the seeds with white outer hulls were washed in running water, sanitized in sodium hypochlorite solution (1% *w*/*w*) for 15 min, and then washed again in running water (Figure 1). Afterward, the seeds were dried in a ventilated oven at 65 °C (±1 °C) for 24 h and cooled to room temperature. Subsequently, to obtain the whole flour, the dried seeds were crushed in an industrial blender; then, the flour was sieved (20 mesh) without residues. The flour was then packaged in polypropylene packaging and stored frozen in a conventional freezer at −18 °C (±1 °C) for subsequent analysis.

### 2.3. Characterization of PSF

For proximate composition, analyses were carried out in triplicate and expressed as percentages [10]. The moisture was measured via oven (Stove for drying and sterilization-393/I, TECNAL, Rio Grande do Norte, Brazil) heating at 105 °C (±2 °C) for 24 h; the ash contents were determined via calcination in a muffle furnace at 550 °C for approximately 6 h until grayish-white ashes were obtained. Proteins were determined from total nitrogen using the Kjeldahl method, requiring the conversion of nitrogen in proteins by a factor of 5.75. Lipids were determined using the Soxhlet continuum extraction method, using hexane as a solvent. Carbohydrates were calculated as the difference—the sum of moisture, proteins, lipids, fibers, and ash content being subtracted from 100%. For the calculation of the total caloric value (TCV), the Atwater coefficient was used, which was 4.0 for proteins, 4.0 for carbohydrates, and 9.0 for lipids. Furthermore, the color was determined using a colorimeter (Colorimeter CR-400, Konica Minolta Sensing Americas, Inc., Ramsey, NJ, USA) to assess the *L***a***b** parameters [11]. The enzymatic-gravimetric method was used with a previously degreased sample [11] to determine the total dietary fiber content.

### 2.4. Preparation of Beef Burger

The burgers were divided into three samples: the standard S without addition of PSF (control group), F5 with the addition of 5% (*w*/*w*) of PSF, and F10 with the addition of 10% (*w*/*w*) of PFS. Chuck ground beef, onion paste, garlic paste, PSF, cold water, table salt, and black pepper were used as ingredients to formulate the burger patties, as described in Table 1.

In the preparation of the burgers, all ingredients were separated and weighed. For the first homogenization, ground beef was mixed with the ingredients (onion, salt, black pepper, garlic). Following this initial mixture, PSF was added to the F5 and F10 samples; cold water was then incorporated, which was the second homogenization, aiding in uniformization of the other ingredients. The meat mass obtained was divided into three portions, the first being the standard one (S), with no addition of PSF (control group), and the other two samples having PSF added in the concentrations of 5% (F5) and 10% (F10).

The patties were shaped using a manual mold measuring 11 cm in diameter and 1.5 cm in height for standardization, weighing an average of 56 g, individually packed in polyethylene bags, and frozen in a freezer at −18 °C until the analyses were carried out.

### 2.5. Characterization of the Burgers

For the proximate composition, the analyses were carried out according to the methodology described in Section 2.3.

To analyze the raw burgers’ water holding capacity (WHC), 5 g of each sample was weighed in test tubes and centrifuged at 3600 rpm for 15 min. The WHC was expressed as a percentage of water holding capacity. The WHC was calculated using the following Equation (1) [12]:(1)$$\mathrm{WHC}\;\%=\left(W_{ac}|W_{bc}\right)\times 100$$
where *W*_ac_ is the sample weight after centrifugation, and *W*_bc_ is the sample weight before centrifugation.

The lipid oxidation determination analysis (TBARS) was carried out at 0, 30, and 60 days of production. Approximately 5 g of the cooled burger samples was placed in triplicate, mixed in 25 mL of trichloroacetic acid 7.5% (TCA), manually homogenized for 1 min, and filtered using qualitative filter paper. After filtration, to quantify malonaldehyde, 4 mL of the filtrate was collected and transferred to the test tubes with lids, to which 5 mL of the 2-thiobarbituric acid (TBA) solution and 1 mL of TCA 7.5% were added. All tubes were homogenized and kept in a hot water bath (90 °C) for 40 min; afterward, they were cooled to room temperature for absorbance reading. To this end, an aliquot was collected for absorbance reading at 538 nm for the spectrophotometer. The TBA values were expressed in mg MDA per kg of sample by means of the standard curve using 1,1,3,3-tetratoxypropane (TEP) [13].

The burger extracts were evaluated for total phenolic content [14] using 20 µL of the extract with 100 µL of the Folin–Ciocalteu reagent diluted in water (1:10). After 3 min, 80 µL of the sodium carbonate solution (75 g/L) was added. After 2 h at room temperature (25 °C), the samples’ absorbances were read in a spectrophotometer (BioTek μQuant Biospectro, Winooski, VT, USA) at 765 nm. A standard curve of gallic acid (10–60 mg/L) was used to calculate the results expressed in mg of gallic acid equivalent per 1 g of dry extract (mg GAE/g).

### 2.6. Characterization of the Burger after Cooking

For the determination of yield and weight loss, the burgers were defrosted at 4 °C for 12 h. The raw burgers were weighed and baked in an electric stove at 180 °C for 30 min until they reached an internal temperature of 72 °C, which was controlled using a thermometer. After baking, they were weighed again [10]. The results were calculated according to Equation (2).
(2)$$\%yield=\frac{weight\;of\;the\;cooked\;sample}{weight\;of\;the\;raw\;sample}\times 100$$

The shrinkage percentage was calculated with a manual caliper measurement of raw burgers and cooked burgers [10]. The results were calculated according to Equation (3):(3)$$\%shrinkage=\frac{\left(diameter\;of\;the\;raw\;sample-diameter\;of\;the\;cooked\;sample\right)}{diameter\;of\;the\;raw\;sample}\times 100$$

Next, a texture analysis (TPA) was performed [12]. After cooking the burgers, the samples were cooled at room temperature. The analyses were carried out using a Texture Analyzer (Brookfield model CT3) for texture profile analysis at 70% (TPA70). To determine TPA70, the burgers were compressed to 70% of their original height using a cylindrical probe with a diameter of 25 mm (P/25) and a speed of 5 mm/s through a two-cycle sequence. The parameters of hardness, cohesiveness, elasticity, chewability, and adhesion were evaluated.

### 2.7. Microbiological Analysis of the Burgers

The microbiological analyses were carried out according to the ANVISA by Normative Instruction no. 161 from 1 July 2022, which establishes microbiological standards for food [15]. Microbiological analyses of burger samples were carried out for treatments on days 0, 30, and 60 of below-freezing storage at −18 °C. Determinations were made using the rapid method 3M™ Petrifilm™ system (3M do Brasil Sumaré/SP).

The burger samples (10 g) were manually homogenized in a sterile bag in 90 mL of sterile peptone water. Then, dilutions were made up to 10^−4^ in sterile peptone solution for subsequent plating on 3M™ Petrifilm™ Count Plaque. Quantification (CFU/g) for *Escherichia coli* was performed on a 3M™ Petrifilm™ *E. coli*/Coliform Count Plate containing modified violet red bile medium and a glucuronidase activity indicator after incubation at 42 ± 1 °C for 24 ± 2 h with blue *E. coli* colonies. Quantification (CFU/g) for positive *Staphylococcus*-coagulase was performed in a 3M™ Petrifilm™ STX Staph Express containing Baird Parker’s modified chromogenic medium, which, after incubation at 37 °C for 24 h, produced red-violet colonies of coagulase-positive *Staphylococcus*. Quantification of mesophilic aerobic bacteria was carried out on 3M™ Petrifilm™ Aerobic Count Plates containing modified Standard Methods nutrients and an indicator for the enumeration of total aerobic bacteria (red colonies) visible after incubation at 35 °C for 48 h. For *Salmonella* sp. presumptive results (presence or absence), burger samples (25 g) were incubated at 41 °C for 24 h in 3M™ *Salmonella* enrichment base medium supplemented with 3M™ *Salmonella* enrichment supplement for subsequent plating on 3M™ Petrifilm™ *Salmonella* Express (SALX) system plates with a chromogenic culture medium system (selective and differential for *Salmonella*). After plate incubation at 41 °C for 48 h, *Salmonella* presented red colonies with yellow zones, sometimes associated with gas bubbles.

### 2.8. Consumer-Targeted Sensory Quality of the Burgers

This research was submitted to the University Federal de Pernambuco, Recife-PE, Brazil Research Ethics Committee, according to resolution CNS 466/2012 regulatory standards for research involving human beings, under opinion 6.002.731, and approved under CAAE 64793522.4.0000.5208. Consumer-targeted sensory quality assessment was carried out at the Department of Nutrition of the University Federal de Pernambuco. The experiment was conducted at the Food Experimentation and Analysis Laboratory (LEAAL), starting at 8:00 a.m. and ending at 5:00 p.m. The tasting was carried out in individual cabins at 20 °C under artificial light, according to [16]. Burgers’ post-cooking temperature was controlled (kept above 70 °C) using an infrared thermometer. The burgers were made available to the consumers at a consumption temperature of approximately 45 °C and were served on disposable white plates, each coded with three random digits, accompanied by a glass of water, as well as water and salt crackers, which aimed to minimize possible interferences between the samples.

The sample population of volunteers participating in the sensory analysis comprised 100 students from the Federal University of Pernambuco (Recife, PE, Brazil) aged between 18 and 25 years old of both sexes (female and male); all were consumers of meat products. Participants were fully informed, in simple language, regarding the research protocol through a Term of Free and Informed Consent (TFIC) and were guided to fill out the form for sensory acceptability evaluation according to the burger’s overall acceptance, appearance, scent, taste, and texture. The individuals scored their degree of acceptance of the analyzed product using a nine-point hedonic scale, with responses ranging from “Really liked” to “Really disliked” [12].

### 2.9. Evaluation of Total Phenolics and Antioxidants in PSF and Burgers

PSF extracts were evaluated in terms of total phenolic content [14]. A gallic acid standard curve (10–60 mg/L) was used to calculate the results expressed in milligrams of gallic acid equivalent (GAE) per 1 gram of dry extract (mg GAE/g).

Total antioxidant capacity (TAC) was calculated using phosphomolybdenum reduction [17], being expressed in relation to ascorbic acid and calculated using the following Equation (4).
(4)$$TAC\%=\frac{\left(sample\;Absorbance-Absorbance\;control\right)}{\left(ascorbic\;acid-control\;Absorbance\right)}\times 100$$

In the DPPH method [7], 1 mg of the extracts was used and diluted in 1 mL of methanol, of which a serial dilution was carried out at concentrations of 15.625–1000 µg/mL. A total volume of 40 µL of these concentrations was added to 250 µL of DPPH reagent in a 96-well plate. After 30 min of incubation in the dark at room temperature, absorbances were read at 517 nm. The percentage of DPPH radical elimination activity was calculated using Equation (5), where Ac = control absorbance, and Ae = sample absorbance (extract or oil):(5)$$DPPH\;elimination\left(\%\right)=\frac{\left(Ac-Ae\right)}{Ac}\times 100$$

For the ABTS method [8], 20 µL of liquid extracts (10%) was mixed with 1 mL of the ABTS solution and left to rest for 6 min. Then, absorbance at 734 nm was read. The elimination of ABTS radicals was estimated as a percentage after the calculation of radical inhibition, calculated using Equation (6), where Ac = control absorbance, Ae = sample absorbance (extract or oil):(6)$$ABTS\;inhibition(\%)=\frac{\left(Ac-Ae\right)}{Ac}\times 100$$

### 2.10. Statistical Analysis

Analyses were carried out in triplicate, and the results obtained were presented as mean ± deviation. Analysis of variance (ANOVA) was performed, and the means were compared using the Tukey test at a 5% significance level (*p* < 0.05).

For analyses of the burgers’ consumer-targeted sensory quality, statistical significance was determined using one-way ANCOVA with a significance level of 5% (*p* < 0.05); global impression was the main predictive effect, with texture, aroma, and color as covariates. All data were analyzed using Statistic 8.0.

## 3. Results and Discussion

### 3.1. PSF Characterization

#### 3.1.1. Physical-Chemical Analysis and Proximal Composition

The approximate composition of PSF can be seen in Table 2. In this study, it was observed that the proximal composition of PSF contains proteins 27.33%, lipids 35.54%, carbohydrates 0.68%, total fiber 31.13%, ash 4.01%, moisture 1.31%, and total caloric value (TCV) 431.90 kcal/100 g. Pumpkin seeds contain between 0.95% and 16.84% of fiber, depending on the variety (pumpkin species) and maturation time. *C. moschata* seeds, 30–90 days after anthesis, have 26–38% fiber; however, when the seed is fully mature, the fiber content decreases to 9.21–10.63% [1,4]. Another study carried out with PSF [6] shows levels of protein 32.86%, lipids 43.46%, carbohydrates 12.16%, total fibers 2.57%, ash 3.32%, moisture 5.66%, and TCV 571.22 kcal/100 g. In the results [6], it was observed that every parameter except ash and fibers had values higher than those of the present research. However, these findings cast PSF in a positive light because the studies show characteristics of a nutritionally rich by-product for food applications.

The differences in PSF’s proximate composition can be attributed to various factors, such as the origin of the pumpkin seeds, the specific pumpkin species, the analysis methods, or the processing techniques used. Additional factors, such as sample preparation and storage, can also contribute to the differences in results [5]. For example, the submersion technique used [6] to remove the antinutritional components, as well as the drying time, may have influenced the findings.

The high level of lipids is attributable to the full use of the entire seed without previous extraction of the oil, which has high values of polyunsaturated fatty acids, especially linoleic acid, representing almost 50%. The energetic value is high in the samples, which is corroborated by the elevated lipid content [18].

The protein content of PSF was found to be 27.33%. A similar study found the protein content of pumpkin peel flour to be 11.51% [10], indicating that PSF is a better protein source compared to pumpkin peel flour. Hussain and collaborators [2] report that the protein content in pumpkin is higher in pumpkin seed powder (14.05–39.75%) than in pumpkin flesh powder (1.30–15.50%) and pumpkin peel powder (1.96–23.95%). Pumpkin cake flour was found to contain 9.14% ash [19], while another study found 6.86% ash content in pumpkin peel flour [10]; both studies revealed a high percentage of ash in pumpkin cake flour and pumpkin peel flour compared to the 4.01% ash content of PSF in this study. This result corroborates that obtained by Aydin et al. [20], who reported a result of 3.55% ash in PSF.

The fiber content in PSF from *Cucurbita moschata* has been measured at 25.35% [11], a value similar to the 31.13% fiber content found in this study. This indicates that PSF, regardless of the specific pumpkin variety, is an excellent source of fiber [5]. It is worth noting that *Cucurbita moschata* seed flour contains a higher proportion of insoluble vs. soluble fibers [11].

#### 3.1.2. PSF Coloring

Food color is assessed visually using various color scales or schemes, including the Hunter *L**, *a**, *b** system [11]. In the present study, values of *a** and *b** for red, green, blue, and yellow, respectively, were referenced for perceptual luminosity. In Figure 2, it is possible to visualize the color of PSF obtained in the present study, with values of *L** 59.89, *a** 3.98, and *b** 31.61, demonstrating that the PSF presents a relatively high luminosity (*L**), with a slight tendency toward red (*a**) and a strong tendency toward yellow (*b**), corresponding to a light-yellow color. These results corroborate those obtained by Fortes et al. [11], who reported results of *L** 59.49, *a** 3.47, and *b** 27.87 in oven-dried PSF (*Cucurbita moschata*).

### 3.2. Beef Burger Patties’ Characterization

#### 3.2.1. Proximate Composition and Physical-Chemical Analyses

According to the Technical Regulation of Identity and Quality for Hamburgers, the amounts of proteins, fat, and carbohydrates are within the parameters required by legislation [21]. Table 2 shows that the amounts of proteins, carbohydrates, lipids, and ash are statistically different (*p* < 0.05) only between samples P and F10. The humidity, fiber, and TCV contents show significant differences (*p* < 0.05) between both PSF samples (F5 and F10) and the S sample.

Study [1] analyzed four formulations of beef burgers with added sweet potato peel flour (SPPF) in different proportions: control, 0.75%, 1.5%, and 2.25%. Compared with the PSF study, the SPPF protein content result was higher, with percentages of 24.98%, 22.05%, 21.60%, and 20.94% for the four respective proportions; yet, the SPFF lipid content was found to be lower, with values of 7.65%, 7.02%, 6.69%, and 6.45%. This inter-study difference in protein and lipid contents may be attributable to the different ingredients in the burgers’ formulations, as well as the different percentages of flour that were added.

The conventional burger contains only saturated fat in its lipid composition, but the addition of PSF to burgers increases the insertion of polyunsaturated fatty acids in a partial replacement of the saturated fat in meat. PSF is a good source of polyunsaturated fatty acids (PUFAs), which mainly comprise linoleic acid [22].

As shown in Table 2, the three samples showed significant differences in moisture (*p* < 0.05): as the flour concentration increased, the moisture percentage diminished. On the other hand, in another study, a group of researchers [8] prepared chicken burgers with the addition of amaranth and pumpkin seeds in different proportions (control, AS1: 1% amaranth; AS2: 2% amaranth; PS1: 1% pumpkin seeds; PS2: 2% pumpkin seeds). They found that the cooking yield, fat, and moisture retention increased as the added amaranth and pumpkin seed content increased, showing an improvement in the hamburgers’ texture and juiciness. With the addition of PSF, fiber contents of 0.88% and 1.76% were observed in F5 and F10, respectively, as shown in Table 2, whereas a study of beef burgers with added sweet potato peel flour showed inferior values of total fibers in all four burger formulations [1]. The use of dietary fibers in meat products brings real benefits: fibers improve the texture by increasing the WHC, showing a good yield, and lowering the cost of the formulation [6].

The pH of the three formulations was found to be 5.96, 6.06, and 6.13 for S, F5, and F10, respectively, with a significant difference (*p* < 0.05) being found between samples S and F10. Because the pH of PSF is closer to neutrality (pH 6.46), supplementing the beef burgers with flour promoted a slight increase in the pH of samples F5 and F10. Such a change may have implications for the microbiological patterns of the burgers during storage time, up to 60 days at −18 °C, as some authors report that pH values above 5.8 favor microbial growth [23]. For water activity (*Aw*), the formulations showed no significant difference (*p* < 0.05). TPC and pH are critical factors that influence the stability and safety of food products [24].

#### 3.2.2. Physical Analyses

The burger samples showed yields of 61.81%, 76.79%, 85.72%; weight loss of 38.18%, 23.21%, 14.28%; and shrinkage rates of 7.27%, 4.03%, 2.43% for samples S, F5, and F10, respectively. It was observed that all three formulations showed significant differences (*p* < 0.05) in all three respects. The greatest weight loss, lowest yield, and lowest shrinkage rate were seen in the standard burger (i.e., with no addition of PSF). In fact, the presence of non-digestible carbohydrates in the formulations of meat products can improve the product’s physical characteristics [25].

In a study of beef burgers with added pumpkin peel flour (PPF), the samples with the highest amount of PPF (3% and 4%) had better yields (84.82% and 87.80%) and lower shrinkage rates (17.95% and 16.13%). These results indicate that the addition of PPF increased the burgers’ capacity to retain water after cooking, probably by increasing the dietary fiber content [10].

The insertion of dietary fibers into the formulations of meat products with salt and fat reductions results in lower weight loss during cooking. Meat products’ water retention is a relevant factor in sensory quality because it is essential to the juiciness and tenderness of the product [3,8]. The results for WHC showed statistical differences between the samples: a higher level of protein content in sample F10 caused an increase in WHC. Another factor worth considering is the type of meat cut used during preparation because different cuts have different WHCs [26].

Color, as one of the main parameters indicating food quality, also influences consumer acceptance. Instrumental color measurements (Hunter Lab and CIELAB systems) correspond to visual assessments of color in a similar way to the human eye. This is possible because the reflection of light on the opaque material is detected by the colorimeter on a scale of three elements, *L** (brightness), *a** (green and red), and *b** (blue and yellow), which makes it possible to eliminate the subjectivity in determining color [27]. In the present study, measuring the color of PSF obtained the following results: *L** 59.89, *a** 3.98, *b** 31.61. These suggest a relatively high luminosity (*L**), with a slight tendency toward red (*a**) and a strong tendency toward yellow (*b**), as seen in Figure 2. These results corroborate those obtained by Nörnberg et al. [28], who found values of *L** 59.49, *a** 3.47, and *b** 27.87 for oven-dried PSF (*Cucurbita moschata*). The color parameters can be observed in Table 3.

Regarding the color of raw and cooked burgers, as shown in Table 3, it is observed that there was a significant increase (*p* < 0.05) in the *L** values between raw sample S compared to samples F5 and F10. In the three raw samples, the *a** values showed no statistical differences (*p* < 0.05), whereas the *b** values did (*p* < 0.05). In the cooked samples, the *L** and *a** values followed the same pattern as the raw samples. For the *b** values, samples S and F10 showed statistical differences (*p* < 0.05) with sample F5. The burgers with PSF added showed darker and less reddish tones after cooking compared to the raw burgers, as shown in Figure 3.

The results of the texture analysis parameters for the burgers are described in Table 3. The connection of proteins with carbohydrates forms a gelatinous network, increasing the compression force and hardness of the burgers [29]. Consequently, significant differences (*p* > 0.05) were found in hardness and chewiness among the three samples analyzed.

The burger samples with quinoa flour (15% and 30%) showed significant differences (*p* > 0.05) regarding hardness, elasticity, and cohesiveness; however, they did demonstrate significant differences (*p* > 0.05) in chewiness [30]. Most of these differences among the studies are attributable to the proximate composition of the flours and the formulations of the burgers.

#### 3.2.3. Flour and Burger Bioactivity Analysis

Statistical differences (*p* < 0.05) were observed between the samples in the total antioxidant capacity (TAC) analysis for antioxidant activity. In the ABTS and DPPH analyses, however, the values showed no relevance. The phenolic compounds showed statistical differences (*p* < 0.05) between samples, as displayed in Table 4. The addition of PSF in the burgers elevated the percentage of antioxidant activity capacity, as well as phenolic compounds.

In a study of fish burgers with an added mix using fish oil and mango seed, the results shown for antioxidant activity were also better presented in the TAC analysis, which is directly related to the oxidative stability of food [31]. However, foods can lose their antioxidant components because of processing and storage: during processing, food oxidation can lead to the loss and/or modification of natural antioxidants, mainly due to oxidation, pyrolysis, and hydrolysis [32].

Burgers infused with goji berry powder were found to contain 0.26 mg/kg (260 mg/g) total phenolics, which exceeded the total phenolic content found here for PSF-infused burgers [33]. Phenolic compounds are influenced by the maturation state of the fruit. Food processing with thermic treatments, homogenization, lyophilization—as well as cooking and culinary preparation methods—can also play an important role in the bioavailability of phenolic compounds [34].

#### 3.2.4. Consumer-Targeted Sensory Quality

The consumer-targeted sensory quality was assessed using 100 untrained tasters. All testers were students at the Federal University of Pernambuco (Recife, PE, Brazil) aged between 18 and 25 years old; 64% were female participants and 36% were male. Table 5 describes the results obtained from the consumer-targeted sensory quality for the attributes of texture, flavor, aroma, color, and overall impression.

Standard, F5, and F10 samples were analyzed for the attributes of color, aroma, flavor, texture, and overall impression (Table 5). The standard and F5 formulations showed no significant differences (*p* > 0.05) for the evaluated attributes. However, compared against the other two samples, the F10 formulation showed differences (*p* < 0.05) for most attributes, except color, for which no significant difference was found between F10 and F5 samples. It can be seen that for standard and F5 treatments, the scores ranged around the average of 7, which is equivalent to the “Liked moderately” concept, and for treatment F10, the scores varied around the average of 6, which is equivalent to the “Liked slightly” concept. Even though consumers did not know which formulations were being tested, their written observations in the evaluation forms for sample F10 described the sample as a more fibrous (more resistant texture), dry beef burger with reduced, less noticeable meat intensity. These reports are likely related to lower humidity, as well as a greater quantity of fibers and proteins in sample F10 in relation to S and F5 (Table 1).

A study of beef burgers with added pumpkin peel flour (PPF) found no significant difference among the five burger formulations with different levels of PPF added—F1 standard (0%), F2 (1%), F3 (2%), F4 (3%), and F5 (4%)—regarding the appearance, texture, color, and purchase intention (*p* > 0.05). However, the F1 formulation was more accepted than F5 for flavor, while F1 and F2 were more accepted than F5 in terms of overall impression [10].

Another study [35] obtained the sensory scores for meatballs formulated with different levels of pumpkin seed kernel flour (PSK), comprising five samples (control, 3%, 6%, 9%, and 12%). The research reported higher sensory scores for the control meatballs, with sensory scores decreasing as PSK flour addition increased (*p* > 0.05). However, no significant difference was found between the control meatballs and those with 3% PSK flour in terms of the appearance, tenderness, flavor, and overall acceptability scores (*p* > 0.05). Overall, all sensory scores decreased when the added PSK flour exceeded 3%. This finding may be attributable to the meatballs’ reduced fat content due to the addition of more PSF, as fat plays an important role in meat products’ flavor, juiciness, and tenderness. Additionally, the color and flavor of pumpkin seeds may be responsible for low sensory scores. These negative effects of non-meat ingredients, when used in large quantities, negatively impact sensory attributes.

Table 6 shows the descriptive statistics of the effect on the overall impression of beef burgers formulated, with texture, flavor, aroma, and color as covariates. The parameters obtained by ANCOVA analysis demonstrate that texture, aroma, and color had a significant effect on the overall impression of tasters in all samples tested (*p* < 0.05) with 95% confidence. On the other hand, the parameters obtained by ANCOVA analysis demonstrate that the flavor had a significant effect on the overall impression of the tasters’ evaluation of both the standard and F5 samples (*p* < 0.05) with 95% confidence; however, this covariate had no significant effect on F10 (*p* > 0.05). The literature has reported that the attributes with the most influence on consumers’ acceptance of meat products are color, flavor, and texture [36].

#### 3.2.5. Microbiological Analysis 

The burgers were formulated without the addition of synthetic ingredients, such as preservatives or flavorings. All raw materials used in the study came from organic cultivation. It was observed that the incorporation of PSF in burger formulations did not negatively impact the microbiological quality of the products. During the 60 days of storage at −18 °C, all burger samples showed an absence of *Salmonella*/25 g, coagulase positive *Staphylococcus* (CFU/g), and *E. coli* (CFU/g). However, after 30 days of storage, all samples showed mesophilic aerobes above 3 × 10^6^ UFC/g, which is considered uncountable.

The count performed at 30 days of storage showed an absence of aerobic mesophiles in the S formulation, 8 × 10^2^ CFU/g in the F5 sample, and 3 × 10^3^ CFU/g in the F10 sample. The data obtained showed satisfactory sanitary conditions for human consumption of the product up to 30 days of storage at −18 °C [15]. After 60 days of storage, the burgers (P, F5, and F10) showed countless (>3 × 10^6^ CFU/g) to aerobic mesophiles, rendering the burgers unsuitable for consumption.

## 4. Conclusions

The use of agro-industrial waste or by-products with nutritional value that contributes to the production of healthier food products as a source of macro- and micronutrients also contributes to the United Nations Sustainable Development Goals (SDGs): End hunger, achieve food security and improve nutrition and promote sustainable agriculture (SDG 2), and Ensure sustainable consumption and production patterns (SDG 12). This research demonstrates that it is feasible to incorporate PSF into meat products, which can improve their nutritional quality and maintain good consumer acceptance of the product. It is important to highlight that this is a “clean label” product, without synthetic additives or chemical preservatives. The incorporation of PSF into the burger at a rate of 5% results in a product with functional claims, as its composition contains the presence of dietary fiber, TAC, and phenolic compounds, in addition to not containing harmful contaminants for consumption. The product is viable for up to 30 days of storage at −18 °C. In future research, experimental studies on animals and/or humans may provide valuable evidence of the nutraceutical action of PSF.

## Figures and Tables

**Figure 1 foods-13-01702-f001:**
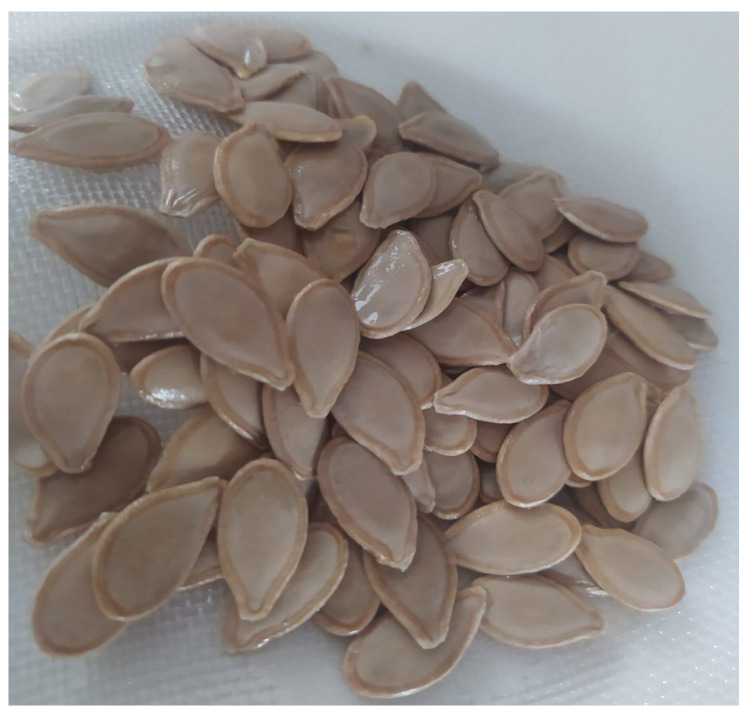
Sanitized pumpkin (*C. moschata* L.) seeds.

**Figure 2 foods-13-01702-f002:**
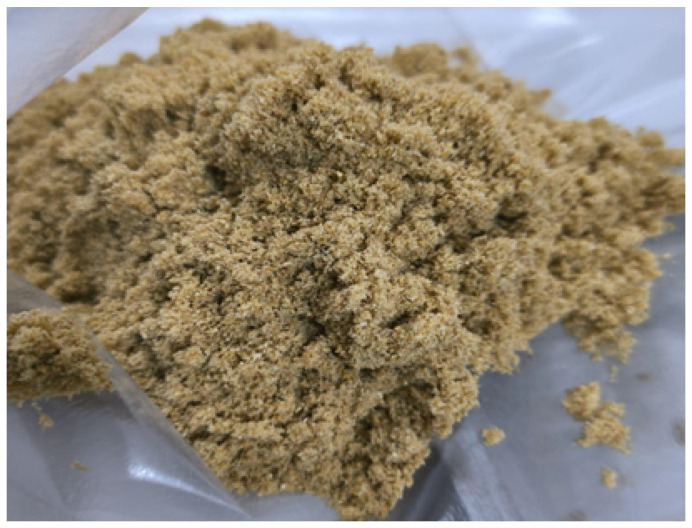
Color of the pumpkin seed flour.

**Figure 3 foods-13-01702-f003:**
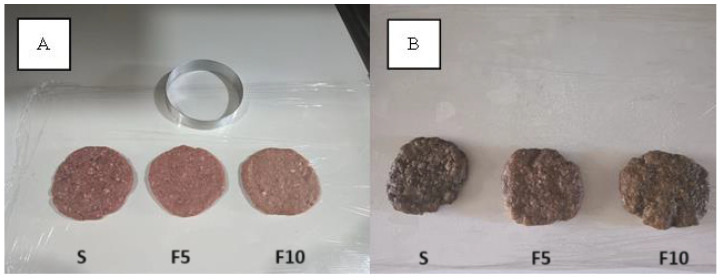
Visual appearance of the raw (**A**) and cooked (**B**) beef burger samples made with (F5, F10) and without (Standard S as control group) the addition of pumpkin seed flour (PSF) in different proportions: 5% (*w*/*w*) (F5) and 10% (*w*/*w*) (F10).

**Table 1 foods-13-01702-t001:** Ingredients and proportions used in the beef burgers’ formulations. “Standard” (control group) indicates no addition of pumpkin seed flour (PSF); “F5” and “F10” indicate the addition of 5% (*w*/*w*) and 10% (*w*/*w*) of PFS, respectively.

Ingredients (%)	Standard	F5	F10
Chuck ground beef	90.0	85.0	80.0
Pumpkin seed flour	0.0	5.0	10.0
Garlic paste	1.5	1.5	1.5
Minced onion	1.5	1.5	1.5
Cold water	5.8	5.8	5.8
Table salt	1.0	1.0	1.0
Black pepper	0.2	0.2	0.2

**Table 2 foods-13-01702-t002:** Proximate compositions of the raw beef burgers made with different pumpkin seed flour (PSF) proportions: no added PSF (Standard) as control group, 5% (*w*/*w*) (F5), and 10% (*w*/*w*) (F10).

Proximate Composition	Standard	F5	F10
Proteins (%)	17.61 ± 0.04 ^a^	18.04 ± 0.41 ^a^	19.86 ± 0.56 ^b^
Carbohydrates (%)	0.63 ± 0.28 ^a^	1.64 ± 0.76 ^a^	2.98 ± 0.63 ^b^
Fat (%)	12.19 ± 0.23 ^a^	12.42 ± 0.20 ^a^	14.55 ± 0.27 ^b^
Moisture (%)	67.79 ± 0.19 ^c^	64.23 ± 0.17 ^b^	58.91 ± 0.16 ^a^
Fibers (%)	0.00 ± 0.00 ^a^	0.88 ± 0.02 ^b^	1.76 ± 0.04 ^c^
TCV (kcal/100 g)	182.71 ± 1.40 ^a^	195.90 ± 3.06 ^b^	221.31 ± 2.06 ^c^

Means in the same row with different letters are significantly different (*p* < 0.05). TCV: Total caloric value.

**Table 3 foods-13-01702-t003:** Instrumental color of the burgers made with and without (Standard, as control group) the addition of pumpkin seed flour (PSF) in different proportions: 5% (*w*/*w*) (F5) and 10% (*w*/*w*) (F10).

Sample	*L**		*a**		*b**	
	Raw	Baked	Raw	Baked	Raw	Baked
Standard	44.92 ± 2.59 ^aA^	42.69 ± 1.91 ^aA^	14.72 ± 1.28 ^aA^	9.09 ± 1.68 ^aB^	14.86 ± 0.94 ^aA^	17.21 ± 0.54 ^aB^
F5	50.56 ± 1.16 ^bA^	48.30 ± 1.54 ^bA^	15.55 ± 0.70 ^aA^	7.03 ± 1.43 ^aB^	16.88 ± 0.37 ^bA^	16.77 ± 1.11 ^aA^
F10	52.74 ± 0.60 ^bA^	50.66 ± 2.22 ^bA^	13.97 ± 1.21 ^aA^	6.40 ± 0.83 ^aB^	19.56 ± 0.78 ^cA^	17.93 ± 0.26 ^aB^

Note: Lowercase letters that are the same vertically and capital letters horizontally do not differ significantly according to the Tukey test with 95% confidence.

**Table 4 foods-13-01702-t004:** Antioxidant activity and phenolic compounds dosage in the pumpkin seed flour (PSF) extract and burger samples. PSF proportions: 5% (*w*/*w*) (F5) and 10% (*w*/*w*) (F10). Standard: without addition of PSF (control group). Phenols.

Sample	TAC %	DPPH %	ABTS %	Phenols *
PSF Extract	17.79 ± 0.64 ^a^	Nd	Nd	142.63 ± 0.18 ^a^
Standard	11.09 ± 0.31 ^c^	3.72 ± 0.14 ^a^	Nd	39.55 ± 0.82 ^c^
F5	18.48 ± 0.51 ^a^	3.07 ± 0.25 ^a^	Nd	82.93 ± 0.32 ^b^
F10	24.45 ± 0.59 ^b^	3.31 ± 0.02 ^a^	Nd	90.30 ± 0.26 ^b^

Note: Values with different letters show a significant difference (*p* < 0.05). Nd: Not determined. * Total phenolic contents mgEAG/g extract.

**Table 5 foods-13-01702-t005:** Consumer-targeted sensory quality of the beef burgers made with different pumpkin seed flour proportions: 5% (*w*/*w*) (F5) and 10% (*w*/*w*) (F10). Standard: without addition of PSF (control group).

Sample	Texture	Flavor	Aroma	Color	Overall Impression
Standard	7.0 ± 1.48 ^a^	7.34 ± 1.32 ^a^	7.10 ± 1.62 ^a^	7.70 ± 1.25 ^a^	7.46 ± 1.30 ^a^
F5	7.42 ± 1.39 ^a^	7.70 ± 1.16 ^a^	7.42 ± 1.26 ^a^	7.28 ± 1.48 ^ab^	7.63 ± 1.20 ^a^
F10	6.28 ± 1.93 ^b^	6.01 ± 2.08 ^b^	6.57 ± 1.59 ^b^	6.83 ± 1.75 ^b^	6.42 ± 1.85 ^b^

Note: Lowercase letters that are the same vertically do not differ significantly using the Tukey test with 95% confidence.

**Table 6 foods-13-01702-t006:** Descriptive statistics of the effect on the overall impression of beef burgers made with different pumpkin seed flour (PSF) proportions—5% (*w*/*w*) (F5) and 10% (*w*/*w*) (F10)—using texture, flavor, aroma, and color as covariates. Standard: without addition of PSF (control group).

**Texture**					
**Sample**	**Mean** **± Standard Deviation**	**SS**	**MS**	**F**	***p* ***
**Standard**	7.46 ± 1.30	109.743	109.743	98.110	0.0000
**F5**	7.63 ± 1.20	321.409	321.409	287.335	0.0000
**F10**	6.42 ± 1.85	13.081	6.541	5.847	0.0032
**Flavor**					
**Sample**	**Mean** **± Standard Deviation**	**SS**	**MS**	**F**	***p* ***
**Standard**	7.46 ± 1.30	62.574	62.574	68.900	0.0000
**F5**	7.63 ± 1.20	383.687	383.689	422.476	0.0000
**F10**	6.42 ± 1.85	0.479	0.239	0.263	0.7685
**Aroma**					
**Sample**	**Mean** **± Standard Deviation**	**SS**	**MS**	**F**	***p* ***
**Standard**	7.46 ± 1.30	110.069	110.069	79.829	0.0000
**F5**	7.63 ± 1.20	244.383	244.383	177.242	0.0000
**F10**	6.42 ± 1.85	31.940	15.970	11.582	0.0000
**Color**					
**Sample**	**Mean** **± Standard Deviation**	**SS**	**MS**	**F**	***p* ***
**Standard**	7.46 ± 1.30	129.273	129.273	83.919	0.0000
**F5**	7.63 ± 1.20	196.535	196.535	127.583	0.0000
**F10**	6.42 ± 1.85	46.369	23.185	15.051	0.0000

***** Values of *p* > 0.05 indicate a non-significant effect.

## Data Availability

The original contributions presented in the study are included in the article, further inquiries can be directed to the corresponding author.

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
