# Peer review of "Development and Evaluation of Nutritional and Quality Standard of Beef Burger Supplemented with Pumpkin (Cucurbita moschata) Seed Flour"

_foods, 2024, doi:10.3390/foods13111702_

Round 1

Reviewer 1 Report

Comments and Suggestions for Authors

Title: USE OF PUMPKIN (Cucurbita moschata) SEEDS IN THE PREPARATION OF FUNCTIONAL BEEF BURGERS

Manuscript Number: foods-2992591

 Above manuscript is about the incorporation of pumpkin seed powder in beef burger to see its effect on physical, chemical, nutritional, functional properties. It is an interesting study to incorporate plant materials in meat products to increase its biological values, increase functional properties, reduce cost etc. Authors did a lot of examinations and analysis to come to the conclusion; however, there are many issues these should be first addressed before it is in Foods.

Title: Title should be comprehensive with the purpose of adding pumpkin seeds in beef burger.

Introduction: What is the purpose of adding pumpkin seeds in beef patties formulation? Please specify it clearly.

Line 46-47: “Besides that, it doesn’t contain gluten, which means it can be recommended for the manufacture of foods that can benefit patients with gluten intolerance or celiac disease [6,7].” Does this statement have any relation with the present work?

Section 2.2. (Obtaining the flour) and Figure 1 and 2: Did you use the seeds with or without outside skin layer? From Fig. 1 and 2, it seems that the authors used the seeds with outer skin layer, which may pose a question of its toxicity or any other harmful effects from the skin layer!

Line 75: Figure 1 probably should be Figure 2 (It is mentioned that the seeds has been already subjected to grind in an industrial blender?)

Table 1: The values should be written properly – instead of using a comma (,), please use a dot (.) for each value. This also applies to Table 2, 3, 4 and 5.

Line 115: Figure 1 is not about beef patties? Please use the appropriate Figure no.

Line 124: Reference [13] – Please provide the details about this reference (Journal name, Volume, Issue, Page etc.). Also provide the formula to calculate WHC.

Section 2.5. Characterization of the “in natura” burger: What is “in natura”?

Line 142: Please provide the calculation formula / method for sodium and potassium.

Line 156-157: Please write the formula using “Insert formula” function in Microsoft Word. Same comment for Line 199-200; line 206-207; and line 211-212.

Section 2.7 (Microbiological analysis): please provide the details about the storage conditions (packaging, temperature etc.) for up to 60 days.

Line 189: Please check the temperature of -45 oC? (The authors mentioned as consumption temperature?)

Line 222 (total fiber 31.13%) Vs. Line 225 (total fiber 2.57%) – what could be the possible reason for such huge difference?

Irrelevant discussion – chickpea protein (line 240), ash content in wheat flour (line 243), rice flour (line 244)?

Section 3.2.1 (PSF coloring): Please italicize L*, a* and b* (whole manuscript). Also, please re-write this section to give the proper meaning in terms of English language and grammar. The present writing very redundant.

Line 328 and 335: Table 2 should be Table 3. Please check!

Line 392-393: 64% female + 34% male participants = 98%; where is remaining 2%?

Line 406: p<0.05 should be p>0.05. Please correct it.

Section 3.2.5 (Microbiological analysis): Please explain this section in a proper way in terms of English grammar and writing as well as in terms of its technical content with discussion of the results. For example, time 30, absence of Salmonella in 25 g, MNP/g Vs. CFU/g etc.

English writing and grammar: The whole manuscript needs thorough revision from a professional manuscript writer in terms of its English language part and grammar. Also, many technical blunders (Latin names, p value, L*, a* and b* values, e instead of and, etc. etc.)

Comments on the Quality of English Language

Moderate English language polishing is necessary.

Author Response

Responses to Reviewers’ Comments

Reviewer #1:

  1. “Title should be comprehensive with the purpose of adding pumpkin seeds in beef burger.”

We appreciate this suggestion, and as requested, the title was amended to “Development and evaluation of nutritional and quality standard of beef burger supplemented with pumpkin (Cucurbita moschata) seed flour.”

  1. Introduction: What is the purpose of adding pumpkin seeds in beef patties formulation? Please specify it clearly.

The authors thank you for the question and have rewritten the INTRODUCTION in order to resolve your concerns. We hope that the changes made have provided the necessary information to clarify the purpose of supplementing beef burgers with pumpkin seed flour. Changes to the text are highlighted in yellow.

  1. Lines 46–47: “Besides that, it doesn’t contain gluten, which means it can be recommended for the manufacture of foods that can benefit patients with gluten intolerance or celiac disease [6,7].” Does this statement have any relation with the present work?

We thank you for the comments made, and we made the adjustments to the text. The INTRODUCTION has been rewritten, and new information has been added with information about functional claims, as well as possible harmful effects or toxicity. Changes to the text are highlighted in yellow.

  1. Section 2.2. (Obtaining the flour) and Figures 1 and 2: Did you use the seeds with or without outside skin layer? From Figs. 1 and 2, it seems that the authors used the seeds with outer skin layer, which may pose a question of its toxicity or any other harmful effects from the skin layer!

Yes, the entire pumpkin seeds were used, containing the white outer hull, also known as the outside skin layer. The relevant information was added to the text, highlighted in yellow, in 2.2. Obtaining the flour (PSF).

  1. Line 75: Figure 1 probably should be Figure 2 (It is mentioned that the seeds has been already subjected to grind in an industrial blender?)

We would like to thank you for checking the correct location for the call-out in Figure 1, which is the photo of the seeds after washing and sanitizing. The change in the text is highlighted in yellow.

  1. Table 1: The values should be written properly – instead of using a comma (,), please use a dot (.) for each value. This also applies to Tables 2, 3, 4 and 5.

The authors apologize for the error in using ‘,’ instead of ‘.’ in the results of Tables 1–5, and we thank the reviewers for their careful attention. The appropriate corrections have been made to all tables and are highlighted in yellow.

  1. Line 115: Figure 1 is not about beef patties? Please use the appropriate Figure no.

We would like to thank you for checking the figure call-out in the text. We removed the text with the call-out to Figure 1 (line 115), which is actually Figure 3, because this figure will appear only in the results and discussion.

  1. Line 124: Reference [13] – Please provide the details about this reference (Journal name, Volume, Issue, Page etc.). Also provide the formula to calculate WHC.

The authors are grateful for verifying the lack of information in reference 13 and we would like to apologize for the error. The missing information in reference 13, as well as the equation of WHC, have been added to the text and are highlighted in yellow.

  1. Section 2.5. Characterization of the “in natura” burger: What is “in natura”?

The term “in natura” was removed from the text, as it was used incorrectly by the authors. By using the term “in natura” the authors intended to refer to a product without additives, and which was produced artisanally (not industrially).

  1. Line 142: Please provide the calculation formula / method for sodium and potassium.

The methodology for determining sodium and potassium was removed (lines 136–142) because the authors found that the results for the determination of sodium and potassium are not described in the manuscript.

  1. Lines 156–157: Please write the formula using “Insert formula” function in Microsoft Word. Same comment for lines 199–200; lines 206–207; and lines 211–212.

The changes were made as requested and are highlighted in yellow.

  1. Section 2.7 (Microbiological analysis): please provide the details about the storage conditions (packaging, temperature, etc.) for up to 60 days.

The requested information has been added to the text and is highlighted in yellow.

  1. Line 189: Please check the temperature of -45C? (The authors mentioned as consumption temperature?)

The authors used the approximation symbol ‘~’ for a temperature of 45 ºC. To avoid doubt, the symbol was replaced by the word “approximately,” highlighted in yellow in the text. The stated temperature of the burgers was measured before consumption.

  1. Line 222 (total fiber 31.13%) Vs. Line 225 (total fiber 2.57%) – what could be the possible reason for such huge difference?

Marcel, Chacha, and Ofoedu [25] did not specify the species of pumpkin they used in the research, nor the maturation time, which may explain the difference between our results and theirs.

Additional information about the amount of fiber in pumpkin seeds has been added to the text and is highlighted in yellow. The authors also added information about the pumpkin species, type of cultivation and period of obtaining the seeds in the methodology (2.1. Reception of raw materials)

  1. Irrelevant discussion – chickpea protein (line 240), ash content in wheat flour (line 243), rice flour (line 244)?

The authors removed the discussion (lines 240–244) and added new text comparing the protein and ash content of pumpkin seeds, specifying the species. New text is highlighted in yellow.

  1. Section 3.2.1 (PSF coloring): Please italicize L*, a* and b* (whole manuscript). Also, please re-write this section to give the proper meaning in terms of English language and grammar. The present writing very redundant.

The changes were made throughout the text (italicize L*, a* and b* in entire manuscript), and the entire manuscript was edited in English (certificate attached). However, the authors believe that reviewer 1 was referring to subitem 3.1.2 PSF coloring, rather than 3.2.1. Subitem 3.1.2 has been rewritten.

  1. Lines 328 and 335: Table 2 should be Table 3. Please check!

As requested, the authors will review the numbering of the tables, and proceed with the correction (Table 3). The changes are highlighted in yellow in the text.

  1. Lines 392–393: 64% female + 34% male participants = 98%; where is remaining 2%?

The authors would like to thank the reviewer for the question. It was verified that the sensory analysis was carried out with 64 women and 36 men. Corrections were made to the text and are highlighted in yellow.

  1. Line 406: p < 0.05 should be p > 0.05. Please correct it.

The authors apologize for the error in using ‘<’ instead of ‘>’, and thanks the reviewers for their careful attention. The correction was made and is highlighted in yellow in the text.

  1. Section 3.2.5 (Microbiological analysis): Please explain this section in a proper way in terms of English grammar and writing as well as in terms of its technical content with discussion of the results. For example, time 30, absence of Salmonella in 25 g, MNP/g vs. CFU/g, etc.

The authors are grateful for comments and recommendations and have made changes in the sections 2.7. Microbiological analysis of the burgers and 3.2.5 Microbiological analysis. The changes made are highlighted in yellow in the text.

  1. English writing and grammar: The whole manuscript needs thorough revision from a professional manuscript writer in terms of its English language part and grammar. Also, many technical blunders (Latin names, p value, L*, a* and b* values, e instead of and, etc. etc.)

The authors submitted the manuscript for editing in English by a professional publisher, and we have revised the manuscript for technical aspects (Latin names, p value, L*, a* and b* values, e instead of and, etc.).

Reviewer 2 Report

Comments and Suggestions for Authors

The paper “USE OF PUMPKIN (Cucurbita moschata) SEEDS IN THE  PREPARATION OF FUNCTIONAL BEEF BURGERS” contributes to the growth of literature for research and food producers, especially nutrient-enriched food with plant products.

Before  the manuscript in “Foods”, the following items should be revised:

Introduction

The description of the aim is not specific. The authors did not formulate a research hypothesis.   "functional claims" - which

I suggest increasing the description of the beneficial properties of pumpkin seed flour. Are there any contraindications?

Methods

Sensory analysis

The methods were adequately described. However, I would suggest to write:

Is the analysis based on a sample of selected consumers or from all over the country?

The group of consumers is small in number (100 people) and therefore the characteristics of this group are very important - age, region.

What was the selection of the group?

Line 392-393 “The sensory analysis was carried out with 100 untrained tasters, where 64% were female participants and 34% were male ones.”  I suggest moving to methodology.

Were the sensory tests conducted in a sensory analysis laboratory equipped with individual booths (at controlled temperature and combined natural/artificial light), designed according to the ISO standard?

What hours were the tests carried out?

 Statistical analysis

(p<0.05) - it should be Italic - p<0.05  - the same in the next ones

 Results

 9-points hedonic scale - this is a scale for consumer assessment, therefore I suggest a change "Sensory evaluation" to “Sensory evaluation among consumers" or “Consumer-Targeted Sensory Quality”, especially to Table 5

 Line 416

“The attributes that have the biggest influence in consumers' acceptance of meat products are color, flavor and texture [55]”

 I suggest that the authors calculate the Statistical significance of predictors of covariance models for OVERALL IMPRESSION and TEXTURE, FLAVOR, AROMA and COLOR. It will increase the scientific level of the results.

 The Discussion

More discussions that are supported with relevant references should be added. 

 Conclusions

The conclusions are extensive, similar to a summary, and missing two summarising sentences. What are the positive effects of the research? What are this research's limitations?

Author Response

REVIEWER 2

INTRODUCTION

  1. The description of the aim is not specific. The authors did not formulate a research hypothesis. "functional claims" – which; I suggest increasing the description of the beneficial properties of pumpkin seed flour. Are there any contraindications?

The authors are grateful for comments and recommendations and have correspondingly rewritten the INTRODUCTION. We hope that the changes made have provided the requested information and that they clarify the reason for supplementing beef burgers with pumpkin seed flour. Changes to the text are highlighted in yellow.

METHODS

  1. Sensory analysis

The methods were adequately described. However, I would suggest to write:

Is the analysis based on a sample of selected consumers or from all over the country? The group of consumers is small in number (100 people) and therefore the characteristics of this group are very important -age, region.

The authors are grateful for the reviewer’s questions, and we have made appropriate changes to the text. Additional information was added regarding the characteristics of the population sample referring to subitem 2.8. Sensory evaluation of the burgers. Changes to the text are highlighted in yellow. The population sample for sensory analysis is local, comprising students and professors at the Federal University of Pernambuco (Recife, PE, Brazil). The sample size followed the standard practices for this type of study (i.e., sensory analysis), which generally ranges between 90 and 150 participants.

  1. What was the selection of the group?

Lines 392–393 “The sensory analysis was carried out with 100 untrained tasters, where 64% were female participants and 34% were male ones.” I suggest moving to methodology.

The authors are grateful for the comments and suggestions made. Information on the percentage of tasters in relation to “sex” (female and male) was only described in the RESULTS, subitem 3.2.4. Sensory analysis, because we had no way of knowing how the sample would be distributed in relation to this parameter. The sample selection criteria were age (18–25) and being students at UFPE. Information that was added to the Methodology, as previously requested.

  1. Were the sensory tests conducted in a sensory analysis laboratory equipped with individual booths (at controlled temperature and combined natural/artificial light), designed according to the ISO standard?

What hours were the tests carried out?

The authors are grateful for the reviewer’s observations and questions. Our changes made in response are highlighted in yellow.

  1. Statistical analysis

(p < 0.05) - it should be Italic - p < 0.05 - the same in the next ones

The authors apologize for the error in using "p" instead of "p" , and thanks the reviewers for their careful attention. Corrections have been made throughout the text and is highlighted in yellow.

RESULTS AND DISCUSSIONS

  1. Nine-point hedonic scale: This is a scale for consumer assessment; therefore, I suggest changing "Sensory evaluation" to “Sensory evaluation among consumers" or “Consumer-targeted sensory quality,” especially for Table 5.

The authors are grateful for the reviewer’s comments and suggestions. The term “Sensory analysis,” when referring to the results of this research, was changed to “Consumer-targeted sensory quality,” including in Table 5, as suggested. The changes are highlighted in yellow.

  1. Line 416: “The attributes that have the biggest influence in consumers' acceptance of meat products are color, flavor and texture [55]”

I suggest that the authors calculate the statistical significance of predictors of covariance models for OVERALL IMPRESSION and TEXTURE, FLAVOR, AROMA and COLOR. It will increase the scientific level of the results.

The authors thank you for the suggestion. However, the sentence cited concerns a report by the authors of the reference item [55]. As suggested, the authors performed descriptive statistics, by ANCOVA, of the effect on the overall impression of beef burgers formulated with texture, flavor, aroma, and color as covariates. Therefore, a new table (Table 6) was included, and new information was added to the text in sections 2.10. Statistical analysis and 3.2.4. Consumer-targeted sensory quality. The changes are highlighted in yellow.

  1. The Discussion: More discussions that are supported with relevant references should be added.

The authors are grateful for the reviewer’s observations. As suggested, we have added new references while removing less relevant references, alongside corresponding changes to the discussion. The changes are highlighted in yellow.

CONCLUSIONS

  1. The conclusions are extensive, similar to a summary, and missing two summarising sentences. What are the positive effects of the research? What are this research's limitations?

The authors are grateful for the comments and suggestions made. The suggested changes were made in CONCLUSIONS and are highlighted in yellow in the text.

Round 2

Reviewer 1 Report

Comments and Suggestions for Authors

Title: DEVELOPMENT AND EVALUATION OF NUTRITIONAL AND QUALITY STANDARD OF BEEF BURGER SUPPLE MENTED WITH PUMPKIN (Cucurbita moschata) SEED FLOUR

Manuscript Number: foods-2992591

The authors have addressed all the issues raised during previous round of reviewing process except following two points. After correcting these two issues, there is no issue from my side.

Line 20-21: Scientific name should be italic!

Table 6: Values not written properly; full stop instead of comma!

Author Response

Author's Reply to the Review Report (Reviewer 1)_Foods-2992591_v2

Recife, 24th May, 2024

The authors have addressed all the issues raised during previous round of reviewing process except following two points. After correcting these two issues, there is no issue from my side.

  1. Line 20-21: Scientific name should be italic!
  2. Table 6: Values not written properly; full stop instead of comma!

The authors would like to thank the reviewer for all his efforts in helping to improve the quality of the manuscript. We would also like to apologize for the errors that remained regarding the use of "," instead of "." in Table 6 and because the scientific name of the pumpkin is not in italics in the Summary (lines 20-21). The requested corrections were made and are highlighted in green in the text (foods-2992591_v2), so as not to be confused with the changes previously requested by reviewers (foods-2992591_v1).   We remain at your disposal for any additional adjustments that may be necessary.

Yours sincerely,

Thayza Stamford
